# REAL-TIME VARIATIONAL METHOD FOR LEARNING NEURAL TRAJECTORY AND ITS DYNAMICS

**Matthew Dowling**
Stony Brook University, New York, USA
`matthew.dowling@stonybrook.edu`

**Yuan Zhao**
National Institute of Mental Health, USA
`yuan.zhao@nih.gov`

**Il Memming Park**
Champalimaud Research, Champalimaud Foundation, Portugal
`memming.park@research.fchampalimaud.org`

## ABSTRACT

Latent variable models have become instrumental in computational neuroscience for reasoning about neural computation. This has fostered the development of powerful offline algorithms for extracting latent neural trajectories from neural recordings. However, despite the potential of real time alternatives to give immediate feedback to experimentalists, and enhance experimental design, they have received markedly less attention. In this work, we introduce the exponential family variational Kalman filter (eVKF), an online recursive Bayesian method aimed at inferring latent trajectories while simultaneously learning the dynamical system generating them. eVKF works for arbitrary likelihoods and utilizes the constant base measure exponential family to model the latent state stochasticity. We derive a closed-form variational analogue to the *predict* step of the Kalman filter which leads to a provably tighter bound on the ELBO compared to another online variational method. We validate our method on synthetic and real-world data, and, notably, show that it achieves competitive performance.

## 1 INTRODUCTION

Population of neurons, especially in higher-order perceptual and motor cortices, show coordinated pattern of activity constrained to an approximately low dimensional 'neural manifold' (Sohn et al., 2019; Churchland et al., 2012; Saxena et al., 2022). The dynamical structure of latent trajectories evolving along the neural manifold is thought to be a valid substrate of neural computation. This idea has fostered extensive experimental studies and the development of computational methods to extract these trajectories directly from electrophysiological recordings. Great strides have been made in developing computational tools for the purpose of extracting latent neural trajectories in *post hoc* neural data analysis.

However, while recently developed tools have proven their efficacy in accurately inferring latent neural trajectories (Pandarinath et al., 2018; Pei et al., 2021; Yu et al., 2009; Zhao & Park, 2017), learning their underlying dynamics has received markedly less attention. Furthermore, even less focus has been placed on real-time methods that allow for online learning of neural trajectories and their underlying dynamics. Real-time learning of neural dynamics would facilitate more efficient experimental design, and increase the capability of closed-loop systems where an accurate picture of the dynamical landscape leads to more precise predictions (Peixoto et al., 2021; Bolus et al., 2021).

In this work, we consider the problem of inferring latent trajectories while simultaneously learning the dynamical system generating them in an online fashion. We introduce the exponential family variational Kalman filter (eVKF), a novel variational inference scheme that draws inspiration from the 'predict' and 'update' steps used in the classic Kalman filter (Anderson & Moore, 1979). We theoretically justify our variational inference scheme by proving it leads to a tighter 'filtering' evidence lower bound (ELBO) than a 'single step' approximation that utilizes the closed form solution of the proposed 'variational prediction' step. Finally, we show how parameterization of the dynamics

via a universal function approximator in tandem with exponential family properties facilitates an alternative optimization procedure for learning the generative model.

Our contributions are as follows: (**i**) We propose a novel variational inference scheme for online learning analogous to the predict and update steps of the Kalman filter. (**ii**) We show the variational prediction step offers a closed form solution when we restrict our variational approximations to *constant base measure* exponential families (Theorem 1). (**iii**) We justify our two step procedure by showing that we achieve a tighter bound on the ELBO, when compared to directly finding a variational approximation to the filtering distribution (Theorem 2). (**iv**) We show that when using universal function approximators for modeling the dynamics, we can optimize our model of the dynamics without propagating gradients through the ELBO as is typically done in variational expectation maximization (vEM) or variational autoencoders (VAEs) (Kingma & Welling, 2014).

## 2 BACKGROUND

### 2.1 STATE-SPACE MODELS

In this paper, we consider observations (e.g. neural recordings), $\mathbf{y}_t$, arriving in a sequential fashion. It is assumed these observations depend directly on a latent Markov process (e.g. structured neural dynamics), $\mathbf{z}_t$, allowing us to write the generative model in state-space form:

$$\mathbf{z}_t \mid \mathbf{z}_{t-1} \sim p_{\boldsymbol{\theta}}(\mathbf{z}_t \mid \mathbf{z}_{t-1}) \qquad \text{(latent dynamics model)}$$
$$\mathbf{y}_t \mid \mathbf{z}_t \quad \sim p_{\boldsymbol{\psi}}(\mathbf{y}_t \mid \mathbf{z}_t) \qquad \text{(observation model)}$$

where $\mathbf{z}_t \in \mathbb{R}^L$, $\mathbf{y}_t \in \mathbb{R}^N$, $\boldsymbol{\psi}$ parameterize the observation model, and $\boldsymbol{\theta}$ parameterize the dynamics model. After observing $\mathbf{y}_t$, any statistical quantities of interest related to $\mathbf{z}_t$ can be computed from the filtering distribution, $p(\mathbf{z}_t \mid \mathbf{y}_{1:t})$. Since we are considering a periodically sampled data streaming setting, it is important that we are able to compute $p(\mathbf{z}_t \mid \mathbf{y}_{1:t})$ in a recursive fashion, with constant time and space complexity.

In addition to inferring the filtering distribution over latent states, we will also be interested in learning the dynamics as the (prior) conditional probability distribution, $p_{\boldsymbol{\theta}}(\mathbf{z}_t \mid \mathbf{z}_{t-1})$, which captures the underlying dynamical law that governs the latent state $\mathbf{z}$ and may implement neural computation. Learning the dynamics facilitates higher quality inference of the latent state, accurate forecasting, and generation of new data. In this paper we will be focused mainly on models where the dynamics are non-linear and parameterized by flexible function approximators. For example, we may model the dynamics as $\mathbf{z}_t \mid \mathbf{z}_{t-1} \sim \mathcal{N}(\mathbf{f}_{\boldsymbol{\theta}}(\mathbf{z}_{t-1}), \mathbf{Q})$, with $\mathbf{f}_{\boldsymbol{\theta}} : \mathbb{R}^L \to \mathbb{R}^L$ parameterized by a neural network.

### 2.2 KALMAN FILTER

Before diving into the general case, let's revisit the well-established Kalman filter (Särkkä, 2013). Given linear Gaussian dynamics and observations, the state-space model description is given by

$$p_{\boldsymbol{\theta}}(\mathbf{z}_t \mid \mathbf{z}_{t-1}) = \mathcal{N}(\mathbf{z}_t \mid \mathbf{A}\mathbf{z}_{t-1}, \mathbf{Q}) \qquad \boldsymbol{\theta} = \{\mathbf{A}, \mathbf{Q}\}$$
$$p_{\boldsymbol{\psi}}(\mathbf{y}_t \mid \mathbf{z}_t) = \mathcal{N}(\mathbf{y}_t \mid \mathbf{C}\mathbf{z}_t + \mathbf{b}, \mathbf{R}) \qquad \boldsymbol{\psi} = \{\mathbf{C}, \mathbf{b}, \mathbf{R}\}$$

The Kalman filter recursively computes the Bayes optimal estimate of the latent state $\mathbf{z}_t$. Given the filtering posterior of previous time step, $p(\mathbf{z}_{t-1} \mid \mathbf{y}_{1:t-1}) = \mathcal{N}(\mathbf{m}_{t-1}, \mathbf{P}_{t-1})$, we first *predict* the latent state distribution (a.k.a. the filtering prior) at time $t$

$$\bar{p}(\mathbf{z}_t \mid \mathbf{y}_{1:t-1}) = \mathbb{E}_{p(\mathbf{z}_{t-1}\mid\mathbf{y}_{1:t-1})} \left[ p_{\boldsymbol{\theta}}(\mathbf{z}_t \mid \mathbf{z}_{t-1}) \right] \tag{1}$$

$$= \mathcal{N}(\mathbf{z}_t \mid \mathbf{A}\mathbf{m}_{t-1}, \mathbf{A}\mathbf{P}_{t-1}\mathbf{A}^\top + \mathbf{Q}) \tag{2}$$

Secondly, we *update* our belief of the current state with the observation $\mathbf{y}_t$ by Bayes' rule

$$p(\mathbf{z}_t \mid \mathbf{y}_{1:t}) \propto p(\mathbf{y}_t \mid \mathbf{z}_t) \, \bar{p}(\mathbf{z}_t \mid \mathbf{y}_{1:t-1}) = \mathcal{N}(\mathbf{z}_t \mid \mathbf{m}_t, \mathbf{P}_t) \tag{3}$$

In order to learn the underlying dynamics $\mathbf{A}$, the linear readout $\mathbf{C}$, state noise $\mathbf{Q}$ and observation noise $\mathbf{R}$, the EM algorithm can be employed (Ghahramani & Hinton, 1996). If a calibrated measure of uncertainty over the model parameters is important, then a prior can be placed over those quantities, and approximate Bayesian methods can be used to find the posterior (Barber & Chiappa, 2006). When the dynamics are nonlinear, then approximate Bayesian inference can be used to compute the posterior over latent states (Kamthe et al., 2022; Hernandez et al., 2018; Pandarinath et al., 2018). Note that these methods are for learning the parameters in the offline setting.

# 3 EXPONENTIAL FAMILY VARIATIONAL KALMAN FILTER (EVKF)

When the models are not linear and Gaussian, the filtering prior Eq. (1) and filtering distribution Eq. (3) are often intractable. This is unfortunate since most models of practical interests deviate in one way or another from these linear Gaussian assumptions. Drawing inspiration from the *predict* and *update* procedure for recursive Bayesian estimation, we propose the *exponential family variational Kalman filter* (eVKF), a recursive variational inference procedure for exponential family models that jointly infers latent trajectories and learns their underlying dynamics.

## 3.1 EXPONENTIAL FAMILY DISTRIBUTIONS

We first take time to recall exponential family distributions, as their theoretical properties make them convenient to work with, especially when performing Bayesian inference. An exponential family distribution can be written as

$$p(\mathbf{z}) = h(\mathbf{z}) \exp\left(\boldsymbol{\lambda}^\top t(\mathbf{z}) - A(\boldsymbol{\lambda})\right) \tag{4}$$

where $h(\mathbf{z})$ is the base measure, $\boldsymbol{\lambda}$ is the natural parameter, $t(\mathbf{z})$ is the sufficient statistics, and $A(\boldsymbol{\lambda})$ is the log-partition function (Wainwright & Jordan, 2008). Many widely used distributions reside in the exponential family; a Gaussian distribution, $p(\mathbf{z}) = \mathcal{N}(\mathbf{m}, \mathbf{P})$, for example, has $t(\mathbf{z}) = \begin{bmatrix} \mathbf{z} & \mathbf{z}\mathbf{z}^\top \end{bmatrix}$, $\boldsymbol{\lambda} = \begin{bmatrix} -\frac{1}{2}\mathbf{P}^{-1}\mathbf{m} & -\frac{1}{2}\mathbf{P}^{-1} \end{bmatrix}$ and $h(\mathbf{z}) = (2\pi)^{-L/2}$. Note that the base measure $h$ does not depend on $\mathbf{z}$ for a Gaussian distribution. We hereby call such an exponential family distribution a *constant base measure* if its base measure, $h$, is constant w.r.t. $\mathbf{z}$. This class encapsulates many well known distributions such as the Gaussian, Bernoulli, Beta, and Gamma distributions.

An additional and important fact we use is that, for a *minimal*[1] exponential family distribution, there exists a one-to-one mapping between the natural parameters, $\boldsymbol{\lambda}$, and the mean parameters, $\boldsymbol{\mu} := \mathbb{E}_{p(\mathbf{z})}[t(\mathbf{z})]$. This mapping is given by $\boldsymbol{\mu} = \nabla_{\boldsymbol{\lambda}} A(\boldsymbol{\lambda})$, and its inverse by $\boldsymbol{\lambda} = \nabla_{\boldsymbol{\mu}} \mathbb{E}_{p(\mathbf{z};\boldsymbol{\lambda}(\boldsymbol{\mu}))}[\log p(\mathbf{z};\boldsymbol{\lambda}(\boldsymbol{\mu}))]$, though $\mathbb{E}_{p(\mathbf{z};\boldsymbol{\lambda}(\boldsymbol{\mu}))}[\log p(\mathbf{z};\boldsymbol{\lambda}(\boldsymbol{\mu}))]$ is usually intractable (Seeger, 2005).

If we have a conditional exponential family distribution, $p_{\boldsymbol{\theta}}(\mathbf{z}_t \mid \mathbf{z}_{t-1})$, then the natural parameters of $\mathbf{z}_t \mid \mathbf{z}_{t-1}$ are a function of $\mathbf{z}_{t-1}$. In this case, we can write the conditional density function as

$$p_{\boldsymbol{\theta}}(\mathbf{z}_t \mid \mathbf{z}_{t-1}) = h(\mathbf{z}_t) \exp(\boldsymbol{\lambda}_{\boldsymbol{\theta}}(\mathbf{z}_{t-1})^\top t(\mathbf{z}_t) - A(\boldsymbol{\lambda}_{\boldsymbol{\theta}}(\mathbf{z}_{t-1}))) \tag{5}$$

where $\boldsymbol{\lambda}_{\boldsymbol{\theta}}(\cdot)$ maps $\mathbf{z}_{t-1}$ to the space of valid natural parameters for $\mathbf{z}_t$. This allows us to use expressive natural parameter mappings, while keeping the conditional distribution in the constant base measure exponential family.

Assume that at time $t$, we have an approximation to the filtering distribution, $q(\mathbf{z}_{t-1})$, and that this approximation is a constant base measure exponential family distribution so that

$$p(\mathbf{z}_{t-1} \mid \mathbf{y}_{1:t-1}) \approx q(\mathbf{z}_{t-1}) = h \exp(\boldsymbol{\lambda}^\top t(\mathbf{z}_{t-1}) - A(\boldsymbol{\lambda})) \tag{6}$$

The primary goal of filtering is to efficiently compute a good approximation $q(\mathbf{z}_t)$ of $p(\mathbf{z}_t \mid \mathbf{y}_{1:t})$, the filtering distribution at time $t$. As we will show, following the two-step variational prescription of, *predict* and then *update*, leads to a natural variational inference scheme and a provably tighter ELBO than a typical single-step variational approximation.

## 3.2 VARIATIONAL PREDICTION STEP

Now that we have relaxed the linear Gaussian assumption, the first problem we encounter is computing the predictive distribution (a.k.a. filtering prior)

$$\bar{p}(\mathbf{z}_t \mid \mathbf{y}_{1:t-1}) = \mathbb{E}_{p(\mathbf{z}_{t-1}|\mathbf{y}_{1:t-1})}[p_{\boldsymbol{\theta}}(\mathbf{z}_t \mid \mathbf{z}_{t-1})] \tag{7}$$

This is generally intractable, since the filtering distribution, $p(\mathbf{z}_{t-1} \mid \mathbf{y}_{1:t-1})$, can only be found analytically for simple SSMs. Similar to other online variational methods (Marino et al., 2018; Zhao & Park, 2020; Campbell et al., 2021), we substitute an approximation for the filtering distribution, $q(\mathbf{z}_{t-1}) \approx p(\mathbf{z}_{t-1} \mid \mathbf{y}_{1:t-1})$, and consider

$$\mathbb{E}_{q(\mathbf{z}_{t-1})}[p_{\boldsymbol{\theta}}(\mathbf{z}_t \mid \mathbf{z}_{t-1})] \tag{8}$$

---

[1] minimality means that all sufficient statistics are linearly independent.

Unfortunately, due to the nonlinearity in $p_{\boldsymbol{\theta}}(\mathbf{z}_t \mid \mathbf{z}_{t-1})$, Eq. (8) is still intractable, making further approximation necessary. We begin by considering an approximation, $\bar{q}(\mathbf{z}_t)$, restricted to a minimal exponential family distribution with natural parameter $\bar{\boldsymbol{\lambda}}$, i.e.

$$\mathbb{E}_{q(\mathbf{z}_{t-1})}[p_{\boldsymbol{\theta}}(\mathbf{z}_t \mid \mathbf{z}_{t-1})] \approx \bar{q}(\mathbf{z}_t) = h \exp(\bar{\boldsymbol{\lambda}}^\top t(\mathbf{z}_t) - A(\bar{\boldsymbol{\lambda}})) \tag{9}$$

Taking a variational approach (Hoffman et al., 2013), our goal is to find the natural parameter $\bar{\boldsymbol{\lambda}}$ that minimizes $\mathbb{D}_{\mathrm{KL}}\big(\bar{q}(\mathbf{z}_t)||\mathbb{E}_{q(\mathbf{z}_{t-1})}[p_{\boldsymbol{\theta}}(\mathbf{z}_t \mid \mathbf{z}_{t-1})]\big)$. Since this quantity cannot be minimized directly, we can consider the following upper bound:

$$\mathcal{F} = -\mathcal{H}(\bar{q}(\mathbf{z}_t)) - \mathbb{E}_{\bar{q}(\mathbf{z}_t)}\mathbb{E}_{q(\mathbf{z}_{t-1})}[\log p_{\boldsymbol{\theta}}(\mathbf{z}_t \mid \mathbf{z}_{t-1})] \geq \mathbb{D}_{\mathrm{KL}}\big(\bar{q}(\mathbf{z}_t)||\mathbb{E}_{q(\mathbf{z}_{t-1})}[p(\mathbf{z}_t \mid \mathbf{z}_{t-1})]\big) \tag{10}$$

Rather than minimizing $\mathcal{F}$ with respect to $\bar{\boldsymbol{\lambda}}$ through numerical optimization, if we take $q(\mathbf{z}_{t-1})$, $\bar{q}(\mathbf{z}_t)$, and $p_{\boldsymbol{\theta}}(\mathbf{z}_t \mid \mathbf{z}_{t-1})$ to be in the same *constant base measure exponential family*, then we can show the following theorem which tells us how to compute the $\bar{\boldsymbol{\lambda}}^*$ that minimizes $\mathcal{F}$.

**Theorem 1** (Variational prediction distribution). *If $p_{\boldsymbol{\theta}}(\mathbf{z}_t \mid \mathbf{z}_{t-1})$, $q(\mathbf{z}_{t-1})$, and $\bar{q}(\mathbf{z}_t)$ are chosen to be in the same minimal and constant base measure exponential family distribution, $\mathcal{E}_c$, then $\bar{q}^*(\mathbf{z}_t) = \mathrm{argmin}_{\bar{q} \in \mathcal{E}_c} \mathcal{F}(\bar{q})$ has a closed form solution given by $\bar{q}^*(\mathbf{z}_t)$ with natural parameters, $\bar{\boldsymbol{\lambda}}_{\boldsymbol{\theta}}$*

$$\bar{\boldsymbol{\lambda}}_{\boldsymbol{\theta}} = \mathbb{E}_{q(\mathbf{z}_{t-1})}[\boldsymbol{\lambda}_{\boldsymbol{\theta}}(\mathbf{z}_{t-1})] \tag{11}$$

Eq. (11) demonstrates that the optimal natural parameters of $\bar{q}$ are the expected natural parameters of the prior dynamics under the variational filtering posterior. While $\bar{\boldsymbol{\lambda}}_{\boldsymbol{\theta}}$ cannot be found analytically, computing a Monte-Carlo approximation is simple; we only have to draw samples from $q(\mathbf{z}_{t-1})$ and then pass those samples through $\boldsymbol{\lambda}_{\boldsymbol{\theta}}(\cdot)$. This also reveals a very nice symmetry that exists between closed form conjugate Bayesian updates and variationally inferring the prediction distribution. In the former case we calculate $\mathbb{E}_{q(\mathbf{z}_{t-1})}[p_{\boldsymbol{\theta}}(\mathbf{z}_t \mid \mathbf{z}_{t-1})]$ while in the latter we calculate $\mathbb{E}_{q(\mathbf{z}_{t-1})}[\boldsymbol{\lambda}_{\boldsymbol{\theta}}(\mathbf{z}_{t-1})]$. We summarize the eVKF two-step procedure in Algorithm 1, located in Appendix E.3.

### 3.3 VARIATIONAL UPDATE STEP

Analogous to the Kalman filter, we *update* our belief of the latent state after observing $\mathbf{y}_t$. When the likelihood is conjugate to the filtering prior, we can calculate a Bayesian update in closed form by using $\bar{q}(\mathbf{z}_t)$ as our prior and computing $p(\mathbf{z}_t \mid \mathbf{y}_{1:t}) \approx q(\mathbf{z}_t) \propto p(\mathbf{y}_t \mid \mathbf{z}_t)\bar{q}(\mathbf{z}_t)$ where $q(\mathbf{z}_t)$, with natural parameter $\boldsymbol{\lambda}$, belongs to the same family as $q(\mathbf{z}_{t-1})$. In the absence of conjugacy, we use variational inference to find $q(\mathbf{z}_t)$ by maximizing the evidence lower bound (ELBO)

$$\boldsymbol{\lambda}^* = \underset{\boldsymbol{\lambda}}{\mathrm{argmax}}\, \mathcal{L}_t(\boldsymbol{\lambda}, \boldsymbol{\theta}) = \underset{\boldsymbol{\lambda}}{\mathrm{argmax}}\, \big[\mathbb{E}_{q(\mathbf{z}_t)}[\log p(\mathbf{y}_t \mid \mathbf{z}_t)] - \mathbb{D}_{\mathrm{KL}}(q(\mathbf{z}_t \mid \boldsymbol{\lambda})||\bar{q}(\mathbf{z}_t))\big] \tag{12}$$

If the likelihood happens to be an exponential family family distribution, then one way to maximize Eq. (12) is through conjugate computation variational inference (CVI) (Khan & Lin, 2017). CVI is appealing in this case because it is equivalent to natural gradient descent, and thus converges faster, and conveniently it operates in the natural parameter space that we are already working in.

### 3.4 TIGHT LOWER BOUND BY THE PREDICT-UPDATE PROCEDURE

A natural alternative to the variational *predict* then *update* procedure prescribed is to directly find a variational approximation to the filtering distribution. One way is to substitute $\mathbb{E}_{q(\mathbf{z}_{t-1})}p_{\boldsymbol{\theta}}(\mathbf{z}_t \mid \mathbf{z}_{t-1})$ for $\bar{q}(\mathbf{z}_t)$ into the ELBO earlier (Marino et al., 2018; Zhao & Park, 2020). Further details are provided in Appendix B, but after making this substitution and invoking Jensen's inequality we get the following lower bound on the log-marginal likelihood at time $t$

$$\mathcal{M}_t = \mathbb{E}_{q(\mathbf{z}_t)}[\log p(\mathbf{y}_t \mid \mathbf{z}_t)] - \mathbb{E}_{q(\mathbf{z}_t)}\big[\log q(\mathbf{z}_t) - \mathbb{E}_{q(\mathbf{z}_{t-1})}[\log p(\mathbf{z}_t \mid \mathbf{z}_{t-1})]\big] \tag{13}$$

However, as we prove in Appendix B, this leads to a provably looser bound on the evidence compared to eVKF, as we state in the following theorem.

**Theorem 2** (Tightness of $\mathcal{L}_t$). *If we set*

$$\Delta(q) = \mathcal{L}_t(q) - \mathcal{M}_t(q) \tag{14}$$

*then, we have that*

$$\Delta(q) = \mathbb{E}_{q(\mathbf{z}_{t-1})}[A(\boldsymbol{\lambda}_{\boldsymbol{\theta}}(\mathbf{z}_{t-1}))] - A(\bar{\boldsymbol{\lambda}}_{\boldsymbol{\theta}}) \geq 0. \tag{15}$$

*so that*

$$\log p(\mathbf{y}_t) \geq \mathcal{L}_t(q) \geq \mathcal{M}_t(q) \tag{16}$$

In other words, the bound on the evidence when using the variational *predict* then *update* procedure is always tighter than the one step procedure. Thus, not only do the variational predict then update steps simplify computations, and make leveraging conjugacy possible, they also facilitate a better approximation to the posterior filtering distribution.

## 3.5 LEARNING THE DYNAMICS

Our remaining desiderata is the ability to learn the parameters of the dynamics model $p_{\boldsymbol{\theta}}(\mathbf{z}_t \mid \mathbf{z}_{t-1})$. One way of learning $\boldsymbol{\theta}$, is to use variational expectation maximization; with $\boldsymbol{\lambda}^*$ fixed, we find the $\boldsymbol{\theta}^*$ that maximizes the ELBO

$$\boldsymbol{\theta}^* = \underset{\boldsymbol{\theta}}{\operatorname{argmax}} \ \mathcal{L}(\boldsymbol{\lambda}^*, \boldsymbol{\theta}) \tag{17}$$

$$= \underset{\boldsymbol{\theta}}{\operatorname{argmin}} \ \mathbb{D}_{\mathrm{KL}}\big(q(\mathbf{z}_t; \boldsymbol{\lambda}^*)||\bar{q}_{\boldsymbol{\theta}}(\mathbf{z}_t; \bar{\boldsymbol{\lambda}}_{\boldsymbol{\theta}})\big) \tag{18}$$

This objective may require expensive computation in practice, e.g. the log-determinant and Cholesky decomposition for Gaussian $q$ and $\bar{q}_{\boldsymbol{\theta}}$. However, since we chose $\bar{q}_{\boldsymbol{\theta}}$ and $q$ to be in the same exponential family, then as described in the following Proposition, we can consider the more computationally tractable square loss function as an optimization objective.

**Proposition 1** (Optimal $\boldsymbol{\theta}$). *If the mapping from $\mathbf{z}_{t-1}$ to the natural parameters of $\mathbf{z}_t$, given by $\boldsymbol{\lambda}_{\boldsymbol{\theta}}(\mathbf{z}_{t-1})$, is a universal function approximator with trainable parameters, $\boldsymbol{\theta}$, then setting*

$$\boldsymbol{\theta}^* = \underset{\boldsymbol{\theta}}{\operatorname{argmin}} \ \tfrac{1}{2}||\boldsymbol{\lambda}^* - \bar{\boldsymbol{\lambda}}_{\boldsymbol{\theta}}||^2 \tag{19}$$

*is equivalent to finding $\boldsymbol{\theta}^* = \operatorname{argmax}_{\boldsymbol{\theta}} \mathcal{L}_t(\boldsymbol{\lambda}^*, \boldsymbol{\theta})$.*

The proposition indicates that we find the optimal $\boldsymbol{\theta}^*$ that matches the natural parameters of predictive distribution to that of the filtering distribution. The proof can be found in Appendix C. Empirically, we have found that even for small neural networks, following Eq. (19), works better in practice than directly minimizing the KL term.

## 3.6 CORRECTING FOR THE UNDERESTIMATION OF VARIANCE

It might be illuminating to take a linear and Gaussian dynamical system, and compare the variational approximation of eVKF to the closed form solutions given by Kalman filtering. Given $p_{\boldsymbol{\theta}}(\mathbf{z}_t \mid \mathbf{z}_{t-1}) = \mathcal{N}(\mathbf{z}_t \mid \mathbf{A}\mathbf{z}_{t-1}, \mathbf{Q})$, the mapping from a realization of $\mathbf{z}_{t-1}$ to the natural parameters of $\mathbf{z}_t$ is given by $\boldsymbol{\lambda}_{\boldsymbol{\theta}}(\mathbf{z}_{t-1}) = \begin{bmatrix} -\frac{1}{2}\mathbf{Q}^{-1}\mathbf{A}\mathbf{z}_{t-1} & -\frac{1}{2}\operatorname{vec}(\mathbf{Q}^{-1}) \end{bmatrix}$. With this mapping, we can determine, in closed form, the prediction distribution given by eVKF. Assuming that $q(\mathbf{z}_{t-1}) = \mathcal{N}(\mathbf{z}_{t-1} \mid \mathbf{m}_{t-1}, \mathbf{P}_{t-1})$, we can find the optimal variational prediction distribution by plugging $\boldsymbol{\lambda}_{\boldsymbol{\theta}}(\mathbf{z}_{t-1})$, into Eq. (11) to find

$$\bar{q}(\mathbf{z}_t) = \mathcal{N}(\mathbf{z}_t \mid \mathbf{A}\mathbf{m}_{t-1}, \mathbf{Q}) \tag{20}$$

However, we know that the prediction step of the Kalman filter returns

$$\bar{p}(\mathbf{z}_t) = \mathcal{N}(\mathbf{z}_t \mid \mathbf{A}\mathbf{m}_{t-1}, \mathbf{Q} + \mathbf{A}\mathbf{P}_{t-1}\mathbf{A}^{\top}) \tag{21}$$

Though this issue has been examined when applying VI to time series models, as in Turner & Sahani (2011), it demonstrates that eVKF underestimates the true variance by an amount $\mathbf{A}\mathbf{P}_{t-1}\mathbf{A}^{\top}$. For this example, we see that because the second natural parameter does not depend on at least second order moments of $\mathbf{z}_{t-1}$, the uncertainty provided by $\mathbf{P}_{t-1}$ will not be propagated forward. At least for the linear and Gaussian case, we can correct this with a post-hoc fix by adding $\mathbf{A}\mathbf{P}_{t-1}\mathbf{A}^{\top}$ to the variance of the variational prediction. If we consider nonlinear Gaussian dynamics with $p_{\boldsymbol{\theta}}(\mathbf{z}_t \mid \mathbf{z}_{t-1}) = \mathcal{N}(\mathbf{z}_t \mid \mathbf{m}_{\boldsymbol{\theta}}(\mathbf{z}_{t-1}), \mathbf{Q})$, then there does not exist an exact correction since the true prediction distribution will not be Gaussian. Empirically, we have found that adding an extended Kalman filter (Särkkä, 2013) like correction of $\mathbf{M}_{t-1}\mathbf{P}_{t-1}\mathbf{M}_{t-1}^{\top}$ to the prediction distribution variance, where $\mathbf{M}_{t-1} = \nabla\mathbf{m}_{\boldsymbol{\theta}}(\mathbf{m}_{t-1})$, helps to avoid overconfidence. In the Appendix E.4 we show a case where not including an additional variance term gives unsatisfactory results when dynamical transitions are Gamma distributed.

## 4 RELATED WORKS

Classic recursive Bayesian methods such as the particle filter (PF), extended Kalman filter (EKF), and unscented Kalman filter (UKF) are widely used for online state-estimation (Särkkä, 2013). Typically, these methods assume a known generative model, but unknown parameters can also be learned by including them through expectation maximization (EM), or dual filtering (Haykin, 2002; Wan & Van Der Merwe, 2000; Wan & Nelson, 1997). While the PF can be used to learn the parameters of the dynamics in an online fashion, as in Kantas et al. (2015), it suffers from the well known issue of "weight degeneracy" limiting its applicability to low dimensional systems. While methods from the subspace identification literature are frequently employed to estimate the underlying dynamics in an offline setting, they are often limited to linear systems (Buesing et al., 2012).

Marino et al. (2018) and Zhao & Park (2020)(VJF), in contrast to eVKF, perform a single step approximation each time instant, which leads to a provably looser bound on the ELBO as stated in Theorem 2. Zhao et al. (2022)(SVMC) use particle filtering to infer the filtering distribution and derives a surrogate ELBO for parameter learning, but because of weight degeneracy it is hard to scale this method to higher dimensional SSMs. Campbell et al. (2021)(OVS) use a backward factorization of the joint posterior. Note that it updates the second most recent state with the most recent observation so that it is technically smoothing rather than filtering, furthermore, the computational complexity of this method can be prohibitive in the online setting as is evident from Table 2.

## 5 EXPERIMENTS

### 5.1 SYNTHETIC DATA AND PERFORMANCE MEASURES

We first evaluate and compare eVKF to other online variational methods as well as classic filtering methods using synthetic data. Since the ground truth is available for synthetic examples, we can measure the goodness of inferred latent states and learned dynamical system in reference to the true ones.

To measure the filtering performance, we use the temporal average log density of inferred filtering distribution evaluated at the true state trajectory: $T^{-1} \sum_{t=1}^{T} \log q(\mathbf{Z}_t; \boldsymbol{\lambda}_t^*)$, where $\boldsymbol{\lambda}_t^*$ are the optimal variational parameters of the approximation of the filtering distribution at time $t$.

To assess the learning of the dynamics model, we sample points around the attractor manifold, evolve them one step forward, and calculate the KL divergence to the true dynamics: $S^{-1} \sum_{i=1}^{S} \mathbb{D}_{\mathrm{KL}}\big(p_{\boldsymbol{\theta}^*}(\mathbf{z}_{t+1} \mid \mathbf{Z}_t^i) || p_{\boldsymbol{\theta}}(\mathbf{z}_{t+1} \mid \mathbf{Z}_t^i)\big)$, where $\mathbf{Z}_t^i$ are the perturbed samples around the attractor manifold (e.g. stable limit cycle) of the true dynamics, $p_{\boldsymbol{\theta}^*}$ is the learned distribution over the dynamics, and $p_{\boldsymbol{\theta}}$ is the true distribution over the dynamics. This helps us evaluate the learned dynamics in the vicinity of the attractor where most samples originate from.

The above divergence measures only the local structure of the learned dynamical system. To evaluate the global structure, we employ the Chamfer distance (Wu et al., 2021)

$$\mathbb{D}_{CD}(S_1 || S_2) = |S_1|^{-1} \sum_{\mathbf{x} \in S_1} \min_{\mathbf{y} \in S_2} ||\mathbf{x} - \mathbf{y}||_2 + |S_2|^{-1} \sum_{\mathbf{y} \in S_2} \min_{\mathbf{x} \in S_1} ||\mathbf{y} - \mathbf{x}||_2 \qquad (22)$$

where $S_1$ and $S_2$ are two distinct sets of points. Usually, this metric is used to evaluate the similarity of point clouds. Intuitively, a low Chamfer distance would mean that trajectories from the learned dynamics would generate a manifold (point cloud) close to the true dynamics—a signature that the attractor structure can be generated. Since the Chamfer distance is not symmetric, we symmetrize it as $\mathbb{D}_{CD}(S_1, S_2) = \frac{1}{2}(\mathbb{D}_{CD}(S_1 || S_2) + \mathbb{D}_{CD}(S_2 || S_1))$ and take the logarithm.

**Chaotic recurrent neural network dynamics.** We first evaluate the filtering performance of eVKF. We consider the chaotic recurrent neural network (CRNN) system used in Campbell et al. (2021); Zhao et al. (2022)

$$p_{\boldsymbol{\theta}}(\mathbf{z}_{t+1} \mid \mathbf{z}_t) = \mathcal{N}(\mathbf{z}_{t+1} \mid \mathbf{z}_t + \Delta \tau^{-1}(\gamma \mathbf{W} \tanh(\mathbf{z}_t) - \mathbf{z}_t), \mathbf{Q})$$

and vary the latent dimensionality. Since we restrict ourselves to filtering, we fix the model parameters at their true values. In addition to the online variational methods, we also include classical filtering algorithms: ensemble Kalman filter (enKF) and bootstrap particle filter (BPF) (Douc et al., 2014).

| METHOD | $L = 2$ | $L = 16$ | $L = 32$ | $L = 64$ |
|---|---|---|---|---|
| EVKF (OURS) | $\mathbf{0.047} \pm 6.4\mathrm{e}{-4}$ | $\mathbf{0.150} \pm 5.8\mathrm{e}{-4}$ | $\mathbf{0.250} \pm 1.5\mathrm{e}{-3}$ | $0.450 \pm 5.8\mathrm{e}{-3}$ |
| OVS | $0.103 \pm 6.4\mathrm{e}{-4}$ | $0.178 \pm 5.8\mathrm{e}{-4}$ | $0.302 \pm 1.5\mathrm{e}{-3}$ | $\mathbf{0.323} \pm 1.5\mathrm{e}{-3}$ |
| VJF | $0.105 \pm 2.8\mathrm{e}{-2}$ | $0.288 \pm 4.0\mathrm{e}{-2}$ | $0.400 \pm 1.1\mathrm{e}{-2}$ | $0.711 \pm 4.4\mathrm{e}{-2}$ |
| ENKF (1,000) | $0.115 \pm 3.3\mathrm{e}{-3}$ | $0.437 \pm 6.0\mathrm{e}{-2}$ | $0.619 \pm 8.2\mathrm{e}{-2}$ | $0.620 \pm 2.8\mathrm{e}{-2}$ |
| BPF (10,000) | $\mathbf{0.047} \pm 6.7\mathrm{e}{-4}$ | $0.422 \pm 9.3\mathrm{e}{-3}$ | $0.877 \pm 2.5\mathrm{e}{-2}$ | $1.660 \pm 4.2\mathrm{e}{-2}$ |

Table 1: **RMSEs of state estimation for Chaotic RNN dynamics.** We show the mean $\pm$ one standard deviation (over 10 trials) of latent state RMSEs. The latent dimensionality $L$ varies from 2 up to 64. Those in the parentheses are the size of ensemble and the number of particles.

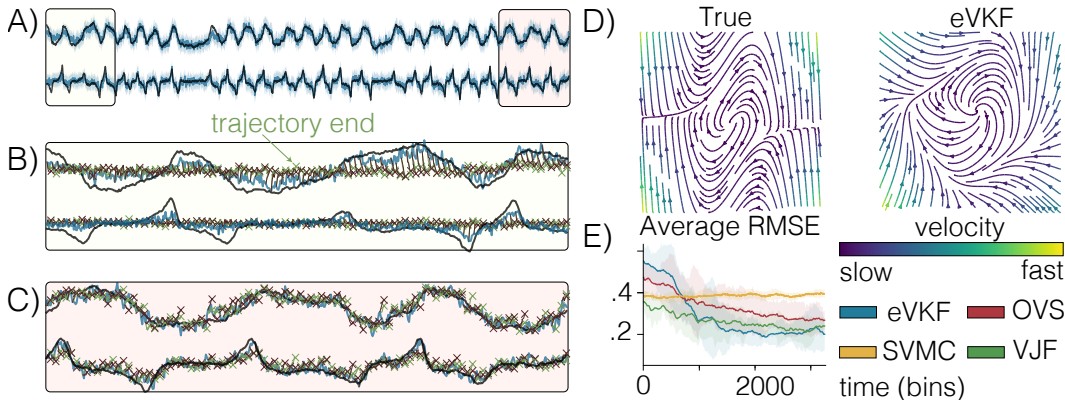

Figure 1: **Van der Pol oscillator with Poisson observations.** **A)** The filtering distribution inferred by eVKF over time, shading indicates the 95% credible interval. **B)** Zoomed in view at the beginning observations. We plot the mean, and trajectories evolved from the filtered mean 5 steps ahead using a "snapshot" of the dynamics at that time, their ending positions are given by the $\times$'s. **C)** Same as before, but at the ending observations. eVKF has learned the dynamics, leading to better filtering capabilities. **D)** True Van der Pol velocity field compared to the dynamics inferred by eVKF. **E)** Moving average RMSE of the filtering mean to the true dynamics, averaged over 10 trials, error bars indicate two standard errors.

Table 1 shows the RMSEs (mean $\pm$ standard deviation over 10 trials of length 250) under increasing latent dimensionality. Surprisingly, eVKF offers competitive performance to the BPF for the 2D case, a regime where the BPF is known to excel. The results show eVKF offers satisfactory results compared to the classic filtering algorithms as well as similar online variational algorithms. We see that OVS performs better in the case $L = 64$, however, this is at the cost of significantly higher computational complexity, as shown in Table 2.

**Learning nonlinear dynamics.** In this experiment we evaluate how well eVKF can learn the dynamics of a nonlinear system that we only have knowledge of through a sequential stream of observations $\mathbf{y}_1, \mathbf{y}_2, \cdots$ and so on. These observations follow a Poisson likelihood with intensity given by a linear readout of the latent state. For the model of the dynamics we consider a noise corrupted Van der Pol oscillator so that the state-space model for this system is given by

$$\mathbf{z}_{t+1,1} = \mathbf{z}_{t,1} + \tfrac{1}{\tau_1}\Delta\mathbf{z}_{t,2} + \sigma\epsilon \qquad \mathbf{z}_{t+1,2} = \mathbf{z}_{t,2} + \tfrac{1}{\tau_2}\Delta(\gamma(1 - \mathbf{z}_{t,1})^2\mathbf{z}_{t,2} - \mathbf{z}_{t,1}) + \sigma\epsilon \qquad (23)$$

$$\mathbf{y}_t \mid \mathbf{z}_t \sim \mathrm{Poisson}(\mathbf{y}_t \mid \Delta\exp(\mathbf{C}\mathbf{z}_t + \mathbf{b})) \qquad (24)$$

where $\exp(\cdot)$ is applied element wise, $\Delta$ is the time bin size, and $\epsilon \sim \mathcal{N}(0,1)$. In order to focus on learning the dynamical system, we fix $\psi = \{\mathbf{C}, \mathbf{b}\}$ at the true values, and randomly initialize the parameters of the dynamics model so that we can evaluate how well eVKF performs filtering and learning the dynamics. We train each method for 3500 data points, freeze the dynamics model, then infer the filtering posterior for 500 subsequent time steps. In Table 2 we report all metrics in addition to the average time per step for both the Poisson and Gaussian likelihood cases. In Figure 1E, we see that eVKF quickly becomes the lowest RMSE filter and remains that way for all 4000 steps.

| METHOD | GAUSSIAN LIKELIHOOD | | | | POISSON LIKELIHOOD | | | |
|---|---|---|---|---|---|---|---|---|
| | $\log q(\mathbf{z}_t)$ ↑ | KL ↓ | $\log($CHAMFER$)$ ↓ | TIME (MS) | $\log q(\mathbf{z}_t)$ ↑ | KL ↓ | $\log($CHAMFER$)$ ↓ | TIME (MS) |
| eVKF | **1.15** | **6.817** | **5.66 $\pm$ 0.93** | 104 | **0.57** | **7.131** | **2.30 $\pm$ 0.32** | **13** |
| OVS | -0.92 | 13.48 | 7.76 $\pm$ 0.16 | 6270 | -0.21 | 9.132 | 3.76 $\pm$ 0.32 | 4150 |
| VJF | -3.58 | 134.3 | 6.61 $\pm$ 0.26 | **30** | -1.24 | 325.5 | 3.99 $\pm$ 0.23 | 100 |
| SVMC | – | 84.83 | 5.85 $\pm$ 0.39 | 314 | – | 410.2 | 4.06 $\pm$ 0.22 | 730 |

Table 2: **Metrics of inference for Van der Pol dynamics**. We report the log-likelihood of the ground truth under the inferred filtering distributions, the KL of one-step transitions, the log symmetric Chamfer distance of trajectories drawn from the learned prior to trajectories realized from the true system, and computation time per time step. SVMC uses 5000 particles.

| METHOD | CONTINUOUS BERNOULLI | | | GAUSSIAN | |
|---|---|---|---|---|---|
| | $\log q(\mathbf{z}_t)$ ↑ | KL ↓ | $\log($CHAMFER$)$ ↓ | $\log q(\mathbf{z}_t)$ ↑ | $\log($CHAMFER$)$ ↓ |
| eVKF | **2.01** | **0.057** | **-0.19 $\pm$ 0.25** | **6.15** | **2.55 $\pm$ 0.36** |
| OVS | – | – | – | 3.61 | 45.74 $\pm$ 16.9 |
| VJF | 1.94 | 2.78 | 3.22 $\pm$ 0.50 | -20.3 | 3.43 $\pm$ 0.13 |
| SVMC (5000) | – | 2.37 | 3.24 $\pm$ 0.45 | – | 3.66 $\pm$ 0.22 |

Table 3: **Metrics of inference for continuous Bernoulli dynamics**. We use both CB and Gaussian approximations for the methods that are applicable. eVKF achieves the highest log-likelihood of latent trajectories, lowest KL-divergence of the learned dynamics, and lowest Chamfer distance. The downside of using Gaussian approximations is most apparent when we look at the Chamfer distance, which is always worse within each method. Note, we do not calculate the KL measure when Gaussian approximations are used.

To examine the computational cost, we report the actual run time per step. Note that OVS took a multi-fold amount of time per step.

**Continuous Bernoulli dynamics.** The constant base measure exponential family opens up interesting possibilities of modeling dynamics beyond additive, independent, Gaussian state noise. Such dynamics could be bounded (i.e. Gamma dynamics) or exist over a compact space (i.e. Beta dynamics). In this example, we consider nonlinear dynamics that are conditionally continuous Bernoulli (CB) (Loaiza-Ganem & Cunningham, 2019) distributed, i.e.

$$p_{\boldsymbol{\theta}}(\mathbf{z}_{t+1} \mid \mathbf{z}_t) = \prod_i \mathcal{CB}(\mathbf{z}_{t+1,i} \mid \mathbf{f}_{\boldsymbol{\theta}}(\mathbf{z}_t)_i) \qquad p(\mathbf{y}_{n,t} \mid \mathbf{z}_t) = \mathcal{N}(\mathbf{y}_{n,t} \mid \mathbf{C}_n^{\top}\mathbf{z}_t, \mathbf{r}_n^2) \qquad (25)$$

where $\mathbf{f}_{\boldsymbol{\theta}} : [0,1]^L \to [0,1]^L$, and $n = 1, \dots, N$. We choose a factorized variational filtering distribution such that $q(\mathbf{z}_t) = \prod_i \mathcal{CB}(\mathbf{z}_{t,i} \mid \boldsymbol{\lambda}_{t,i})$, where $\boldsymbol{\lambda}_{t,i}$ is the $i$-th natural parameter at time $t$. In Fig. 2 we show that eVKF is able to learn an accurate representation of the dynamics underlying the observed data. Fig. 2B also demonstrates that a CB prior over the dynamics is able to generate trajectories much more representative of the true data compared to a Gaussian approximation. These results show CB dynamics could be a proper modeling choice if a priori the dynamics are known to be compact, and exhibit switching like behavior. In Table 3 we report the performance of eVKF and the other methods on synthetic data generated from the state-space model above when using both CB and Gaussian approximations. Notably, we see the Chamfer metric is lower within each method when using CB approximation, showing that even though the true filtering distribution might not exactly be a CB distribution, it is still a good choice.

## 5.2 ELECTROPHYSIOLOGICAL RECORDING DURING A REACHING TASK

To evaluate eVKF with real-world neural data, we considered electrophysiological recordings taken from monkey motor cortex during a reaching task (Churchland et al., 2012). This dataset has typically been used to evaluate latent variable modeling of neural population activity (Pei et al., 2021).

In each trial of the experiment, a target position is presented to the monkey, after which it must wait a randomized amount of time until a "Go" cue, signifying that the monkey should reach toward the target. We first take 250 random trials from the experiment, and use latent states inferred by Gaussian process factor analysis (GPFA) (Yu et al., 2009) to pretrain eVKF's model of the dynamics. Then, we

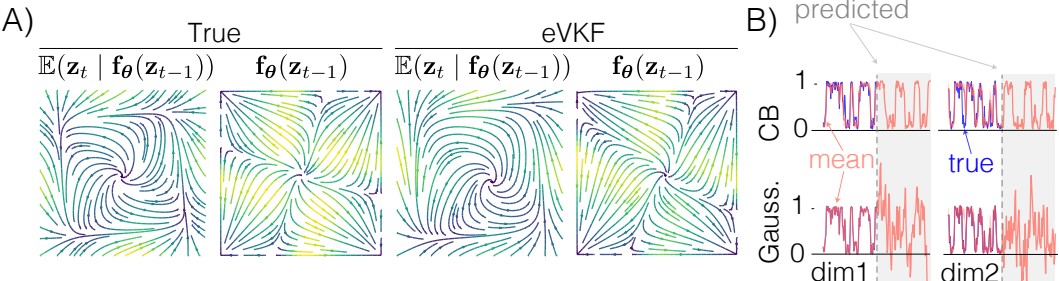

Figure 2: Continuous Bernoulli dynamics. **A)** Velocity field for both $\mathbb{E}(\mathbf{z}_t \mid \mathbf{f}_{\boldsymbol{\theta}}(\mathbf{z}_{t-1}))$ and $\mathbf{f}_{\boldsymbol{\theta}}(\mathbf{z}_{t-1})$ from the synthetically created continuous Bernoulli dynamics, and those inferred by eVKF. We see that in mean there are limit cycle dynamics, but for the states to actually saturate at the boundary there have to be strong attractor dynamics in parameter space. **B)** Inferred filtering distributions when using Gaussian approximations compared to continuous Bernoulli approximations; Gaussian distributions are able to infer the latent state well – but they cannot generate similar trajectories, as we see from trajectories propagated forward through the learned dynamics (shaded in gray)

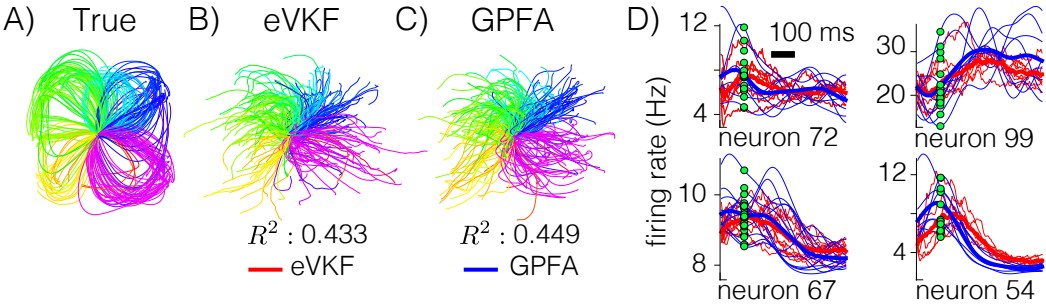

Figure 3: **A)** True hand movements from fixation point to target. **B)** The hand position given by the velocity that we linearly decode using eVKF's inferred firing rates. **C)** Same as previous, but for GPFA. We see that the $R^2$ value, and decoded hand positions using eVKF are competitive with GPFA. **D)** Single trial (thin lines), and condition average (bold lines) firing rates for select neurons and tasks, aligned to the movement onset (demarcated with green dots)

use eVKF to perform filtering and update the dynamics model on a disjoint set of 250 trials. In order to determine if eVKF learns a useful latent representation, we examine if the velocity of the monkey's movement can be linearly decoded using the inferred filtering distribution. In Fig. 3B, we show the decoded hand position from the smoothed firing rates inferred by eVKF in parallel to the result of GPFA. eVKF is able to achieve competitive performance even though GPFA is a smoothing method. In Fig. 3C, we plot the single trial firing rates of some neurons over selected reaching conditions, showing that even for single trials, eVKF can recover firing rates decently.

## 6 CONCLUSION

We tackled the problem of inferring latent trajectories and learning the dynamical system generating them in real-time— for Poisson observation, processing took $\sim 10$ ms per sample. We proposed a novel online recursive variational Bayesian joint filtering method, eVKF, which allows rich and flexible stochastic state transitions from any constant base measure exponential family for arbitrary observation distributions. Our two-step variational procedure is analogous to the Kalman filter, and achieves a tighter bound on the ELBO than the previous methods. We demonstrated that eVKF performs on par with competitive online variational methods of filtering and parameter learning. For future work, we will focus on extensions to the full exponential family of distributions, characterizing the variance lost in more generality, and improving performance as latent dimensionality is scaled up. Future work will also incorporate learning the parameters of the likelihood $\psi$ into eVKF, rather than focusing only on the dynamics model parameters and filtering states.

ACKNOWLEDGEMENTS

MD and IP were supported by an NSF CAREER Award (IIS-1845836) and NIH RF1DA056404. YZ was supported in part by the National Institute of Mental Health Intramural Research Program (ZIC-MH002968). We thank the anonymous reviewers for their helpful feedback and comments, and Josue Nassar for helpful suggestions for improving the manuscript.

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

APPENDIX

## A  PROOF OF THEOREM 1

**Theorem 1** (Variational prediction distribution). *If $p_{\boldsymbol{\theta}}(\mathbf{z}_t \mid \mathbf{z}_{t-1})$, $q(\mathbf{z}_{t-1})$, and $\bar{q}(\mathbf{z}_t)$ are chosen to be in the same minimal and constant base measure exponential family distribution, $\mathcal{E}_c$, then $\bar{q}^*(\mathbf{z}_t) = \operatorname{argmin}_{\bar{q} \in \mathcal{E}_c} \mathcal{F}(\bar{q})$ has a closed form solution given by $\bar{q}^*(\mathbf{z}_t)$ with natural parameters, $\bar{\boldsymbol{\lambda}}_{\boldsymbol{\theta}}$*

$$\bar{\boldsymbol{\lambda}}_{\boldsymbol{\theta}} = \mathbb{E}_{q(\mathbf{z}_{t-1})} \left[ \boldsymbol{\lambda}_{\boldsymbol{\theta}}(\mathbf{z}_{t-1}) \right] \tag{11}$$

**Proof:** The upper bound we want to minimize is given by

$$\mathcal{F} = -\mathcal{H}(\bar{q}(\mathbf{z}_t)) - \mathbb{E}_{\bar{q}(\mathbf{z}_t)} \mathbb{E}_{q(\mathbf{z}_{t-1})} \left[ \log p_{\boldsymbol{\theta}}(\mathbf{z}_t \mid \mathbf{z}_{t-1}) \right] \tag{26}$$

where $\mathcal{H}(\bar{q})$ is the entropy of $\bar{q}$. For exponential family distributions, we recall that the negative entropy coincides with the conjugate dual of the log partition function, or $-\mathcal{H}(q_{\boldsymbol{\mu}}) = A^*(\boldsymbol{\mu})$ (Wainwright & Jordan, 2008). Then, we have that,

$$\mathcal{F} = -\mathbb{E}_{\bar{q}(\mathbf{z}_t)} \mathbb{E}_{q(\mathbf{z}_{t-1})} \left[ \boldsymbol{\lambda}_{\boldsymbol{\theta}}(\mathbf{z}_{t-1})^\top t(\mathbf{z}_t) - A(\boldsymbol{\lambda}(\mathbf{z}_{t-1})) - \log h(\mathbf{z}_t) \right] + A^*(\bar{\boldsymbol{\mu}}) \tag{27}$$

$$= -\mathbb{E}_{\bar{q}(\mathbf{z}_t)} \mathbb{E}_{q(\mathbf{z}_{t-1})} \left[ \boldsymbol{\lambda}_{\boldsymbol{\theta}}(\mathbf{z}_{t-1})^\top t(\mathbf{z}_t) - A(\boldsymbol{\lambda}(\mathbf{z}_{t-1})) - \log h \right] + A^*(\bar{\boldsymbol{\mu}}) \tag{28}$$

$$= -\bar{\boldsymbol{\mu}}^\top \mathbb{E}_{q(\mathbf{z}_{t-1})} \left[ \boldsymbol{\lambda}_{\boldsymbol{\theta}}(\mathbf{z}_{t-1}) \right] + A^*(\bar{\boldsymbol{\mu}}) + \text{constants} \tag{29}$$

$$= -\bar{\boldsymbol{\mu}}^\top \bar{\boldsymbol{\lambda}}_{\boldsymbol{\theta}} + A^*(\bar{\boldsymbol{\mu}}) + \text{constants} \tag{30}$$

where in the first line we use the fact that $\mathbb{E}_{\bar{q}(\mathbf{z}_t)} \left[ t(\mathbf{z}_t) \right] = \bar{\boldsymbol{\mu}}$. In the second line, we use the fact that $p_{\boldsymbol{\theta}}(\mathbf{z}_t \mid \mathbf{z}_{t-1})$ has a constant base measure. In the third line we separate out terms that are constant with respect to $\bar{\boldsymbol{\lambda}}$. In the fourth line we use the definition $\bar{\boldsymbol{\lambda}}_{\boldsymbol{\theta}} := \mathbb{E}_{q(\mathbf{z}_{t-1})} \left[ \boldsymbol{\lambda}_{\boldsymbol{\theta}}(\mathbf{z}_{t-1}) \right]$.

Since our equation is in terms of the mean parameters of $\bar{q}$, thanks to the minimality of $\bar{q}$, we can just consider the optimal variational parameters in their mean parameterization. Then, by considering maximization of $-\mathcal{F}$ rather than minimization of $\mathcal{F}$, we can write that the optimal variational parameters satisfy

$$\bar{\boldsymbol{\mu}}^* = \operatorname*{argmax}_{\bar{\boldsymbol{\mu}}} \left[ \bar{\boldsymbol{\mu}}^\top \bar{\boldsymbol{\lambda}}_{\boldsymbol{\theta}} - A^*(\bar{\boldsymbol{\mu}}) \right] \tag{31}$$

Taking derivatives of the right hand side and setting it equal to 0, we have that

$$\bar{\boldsymbol{\lambda}}_{\boldsymbol{\theta}} - \nabla_{\bar{\boldsymbol{\mu}}^*} A^*(\bar{\boldsymbol{\mu}}^*) = 0 \tag{32}$$

$$\bar{\boldsymbol{\lambda}}_{\boldsymbol{\theta}} - \bar{\boldsymbol{\lambda}}^* = 0 \tag{33}$$

$$\bar{\boldsymbol{\lambda}}^* = \bar{\boldsymbol{\lambda}}_{\boldsymbol{\theta}} \tag{34}$$

where in the second line we use the fact that $\boldsymbol{\lambda} = \nabla_{\boldsymbol{\mu}} A^*(\boldsymbol{\mu})$. As stated in Theorem 1, we have that $\bar{\boldsymbol{\lambda}}^* = \bar{\boldsymbol{\lambda}}_{\boldsymbol{\theta}}$ as claimed.

## B  PROOF OF THEOREM 2

**Theorem 2** (Tightness of $\mathcal{L}_t$). *If we set*

$$\Delta(q) = \mathcal{L}_t(q) - \mathcal{M}_t(q) \tag{14}$$

*then, we have that*

$$\Delta(q) = \mathbb{E}_{q(\mathbf{z}_{t-1})} \left[ A(\boldsymbol{\lambda}_{\boldsymbol{\theta}}(\mathbf{z}_{t-1})) \right] - A(\bar{\boldsymbol{\lambda}}_{\boldsymbol{\theta}}) \geq 0. \tag{15}$$

*so that*

$$\log p(\mathbf{y}_t) \geq \mathcal{L}_t(q) \geq \mathcal{M}_t(q) \tag{16}$$

**Proof:** We write the two ELBOs as

$$\mathcal{L}_t(q) = \mathbb{E}_{q(\mathbf{z}_t)} \log p(\mathbf{y}_t \mid \mathbf{z}_t) - \mathbb{D}_{\mathrm{KL}}(q(\mathbf{z}_t)||\bar{q}(\mathbf{z}_t)) \tag{35}$$

$$\mathcal{M}_t(q) = \mathbb{E}_{q(\mathbf{z}_t)} \log p(\mathbf{y}_t \mid \mathbf{z}_t) - \mathbb{E}_{q(\mathbf{z}_t)} \left[ \log q(\mathbf{z}_t) - \mathbb{E}_{q(\mathbf{z}_{t-1})} \log p(\mathbf{z}_t \mid \mathbf{z}_{t-1}) \right] \tag{36}$$

This means the difference, $\Delta(q) := \mathcal{L}_t(q) - \mathcal{M}_t(q)$, can be written as

$$
\begin{aligned}
\Delta(q) &= \mathbb{E}_{q(\mathbf{z}_t)} \left[ \log \bar{q}(\mathbf{z}_t) - \mathbb{E}_{q(\mathbf{z}_{t-1})} \left[ \log p(\mathbf{z}_t \mid \mathbf{z}_{t-1}) \right] \right] \\
&= \mathbb{E}_{q(\mathbf{z}_t)} \left\{ \log h(\mathbf{z}_t) + \bar{\boldsymbol{\lambda}}_{\boldsymbol{\theta}}^{\top} t(\mathbf{z}_t) - A(\bar{\boldsymbol{\lambda}}_{\boldsymbol{\theta}}) - \mathbb{E}_{q(\mathbf{z}_{t-1})} \left[ \log h(\mathbf{z}_t) + \boldsymbol{\lambda}_{\boldsymbol{\theta}}(\mathbf{z}_{t-1})^{\top} t(\mathbf{z}_t) - A(\boldsymbol{\lambda}_{\boldsymbol{\theta}}(\mathbf{z}_{t-1})) \right] \right\} \\
&= (\bar{\boldsymbol{\lambda}}_{\boldsymbol{\theta}} - \underbrace{\mathbb{E}_{q(\mathbf{z}_{t-1})} [\boldsymbol{\lambda}_{\boldsymbol{\theta}}(\mathbf{z}_{t-1})]}_{\bar{\boldsymbol{\lambda}}_{\boldsymbol{\theta}}})^{\top} \mathbb{E}_{q(\mathbf{z}_t)} [t(\mathbf{z}_t)] + \left( \mathbb{E}_{q(\mathbf{z}_{t-1})} [A(\boldsymbol{\lambda}_{\boldsymbol{\theta}}(\mathbf{z}_{t-1}))] - A(\bar{\boldsymbol{\lambda}}_{\boldsymbol{\theta}}) \right) \\
&= \mathbb{E}_{q(\mathbf{z}_{t-1})} [A(\boldsymbol{\lambda}_{\boldsymbol{\theta}}(\mathbf{z}_{t-1}))] - A(\bar{\boldsymbol{\lambda}}_{\boldsymbol{\theta}})
\end{aligned}
$$

Invoking Jensen's inequality, and the fact that the log-partition function is convex in its arguments, we can write that

$$\mathbb{E}_{q(\mathbf{z}_{t-1})} [A(\boldsymbol{\lambda}_{\boldsymbol{\theta}}(\mathbf{z}_{t-1}))] \geq A(\underbrace{\mathbb{E}_{q(\mathbf{z}_{t-1})} [\boldsymbol{\lambda}_{\boldsymbol{\theta}}(\mathbf{z}_{t-1})]}_{\bar{\boldsymbol{\lambda}}_{\boldsymbol{\theta}}}) \tag{37}$$

which means that

$$\Delta(q) = \mathbb{E}_{q(\mathbf{z}_{t-1})} [A(\boldsymbol{\lambda}_{\boldsymbol{\theta}}(\mathbf{z}_{t-1}))] - A(\bar{\boldsymbol{\lambda}}_{\boldsymbol{\theta}}) \geq 0 \tag{38}$$

as claimed.

## C  PROOF OF PROPOSITION 1

**Proposition 1** (Optimal $\boldsymbol{\theta}$). *If the mapping from $\mathbf{z}_{t-1}$ to the natural parameters of $\mathbf{z}_t$, given by $\boldsymbol{\lambda}_{\boldsymbol{\theta}}(\mathbf{z}_{t-1})$, is a universal function approximator with trainable parameters, $\boldsymbol{\theta}$, then setting*

$$\boldsymbol{\theta}^* = \underset{\boldsymbol{\theta}}{\operatorname{argmin}} \tfrac{1}{2} ||\boldsymbol{\lambda}^* - \bar{\boldsymbol{\lambda}}_{\boldsymbol{\theta}}||^2 \tag{19}$$

*is equivalent to finding $\boldsymbol{\theta}^* = \operatorname{argmax}_{\boldsymbol{\theta}} \mathcal{L}_t(\boldsymbol{\lambda}^*, \boldsymbol{\theta})$.*

**Proof:** By the fact that $\mathcal{L}(\boldsymbol{\lambda}^*, \boldsymbol{\theta}) = \mathbb{E}_{q(\mathbf{z}_t)} \log p(\mathbf{y}_t \mid \mathbf{z}_t) - \mathbb{D}_{\mathrm{KL}}(q(\mathbf{z}_t)||\bar{q}_{\boldsymbol{\theta}}(\mathbf{z}_t))$, we have that $\nabla_{\boldsymbol{\theta}} \mathcal{L}(\boldsymbol{\lambda}^*, \boldsymbol{\theta}) = -\nabla_{\boldsymbol{\theta}} \mathbb{D}_{\mathrm{KL}}(q(\mathbf{z}_t)||\bar{q}_{\boldsymbol{\theta}}(\mathbf{z}_t))$ so that

$$
\begin{aligned}
\nabla_{\boldsymbol{\theta}} \mathbb{D}_{\mathrm{KL}}(q(\mathbf{z}_t)||\bar{q}(\mathbf{z}_t)) &= \nabla_{\boldsymbol{\theta}} \mathbb{E}_{q(\mathbf{z}_t)} \left[ t(\mathbf{z}_t) \left( \boldsymbol{\lambda}^* - \bar{\boldsymbol{\lambda}}_{\boldsymbol{\theta}} \right) + A(\bar{\boldsymbol{\lambda}}_{\boldsymbol{\theta}}) \right] \tag{39} \\
&= -\nabla_{\boldsymbol{\theta}} \bar{\boldsymbol{\lambda}}_{\boldsymbol{\theta}} \boldsymbol{\mu}^* + [\nabla_{\boldsymbol{\theta}} \bar{\boldsymbol{\lambda}}_{\boldsymbol{\theta}}] \nabla_{\bar{\boldsymbol{\lambda}}_{\boldsymbol{\theta}}} A(\bar{\boldsymbol{\lambda}}_{\boldsymbol{\theta}}) \tag{40} \\
&= -\nabla_{\boldsymbol{\theta}} \bar{\boldsymbol{\lambda}}_{\boldsymbol{\theta}} \boldsymbol{\mu}^* + [\nabla_{\boldsymbol{\theta}} \bar{\boldsymbol{\lambda}}_{\boldsymbol{\theta}}] \bar{\boldsymbol{\mu}}_{\boldsymbol{\theta}} \tag{41} \\
&= [\nabla_{\boldsymbol{\theta}} \bar{\boldsymbol{\lambda}}_{\boldsymbol{\theta}}] (\bar{\boldsymbol{\mu}}_{\boldsymbol{\theta}} - \boldsymbol{\mu}^*) \tag{42}
\end{aligned}
$$

whereas, for the alternative objective we have that

$$\nabla_{\boldsymbol{\theta}} \tfrac{1}{2} ||\boldsymbol{\lambda} - \bar{\boldsymbol{\lambda}}_{\boldsymbol{\theta}}||^2 = [\nabla_{\boldsymbol{\theta}} \bar{\boldsymbol{\lambda}}_{\boldsymbol{\theta}}] (\bar{\boldsymbol{\lambda}}_{\boldsymbol{\theta}} - \boldsymbol{\lambda}^*) \tag{43}$$

Assume that $\boldsymbol{\lambda}_{\boldsymbol{\theta}}(\cdot)$ is a flexible enough function approximator so that $[\nabla_{\boldsymbol{\theta}} \bar{\boldsymbol{\lambda}}_{\boldsymbol{\theta}}]$ has full column rank. Then if the gradient of the KL term is 0, either $[\nabla_{\boldsymbol{\theta}} \bar{\boldsymbol{\lambda}}_{\boldsymbol{\theta}}]$ is 0, in which case Eq. (43) is 0, or $(\bar{\boldsymbol{\lambda}}_{\boldsymbol{\theta}} - \boldsymbol{\lambda}^*)$ is 0, which by continuity of the mapping from natural to mean parameters implies that $(\bar{\boldsymbol{\mu}}_{\boldsymbol{\theta}} - \boldsymbol{\mu}^*)$ is 0. Showing equivalence of stationary points of the two objectives.

## D  VARIANCE CORRECTION FOR NONLINEAR GAUSSIAN DYNAMICS

For the case of nonlinear Gaussian dynamics, we could consider directly linearizing the dynamics in order to forego solving a variational problem (e.g. linearizing the dynamics of Eq. 8 about the mean of $\mathbf{z}_{t-1}$ to evaluate the expectation directly). Concretely, consider nonlinear Gaussian dynamics specified via $p(\mathbf{z}_t \mid \mathbf{z}_{t-1}) = \mathcal{N}(\mathbf{z}_t \mid \mathbf{m}_{\boldsymbol{\theta}}(\mathbf{z}_{t-1}), \mathbf{Q})$; assuming we have a variational approximation

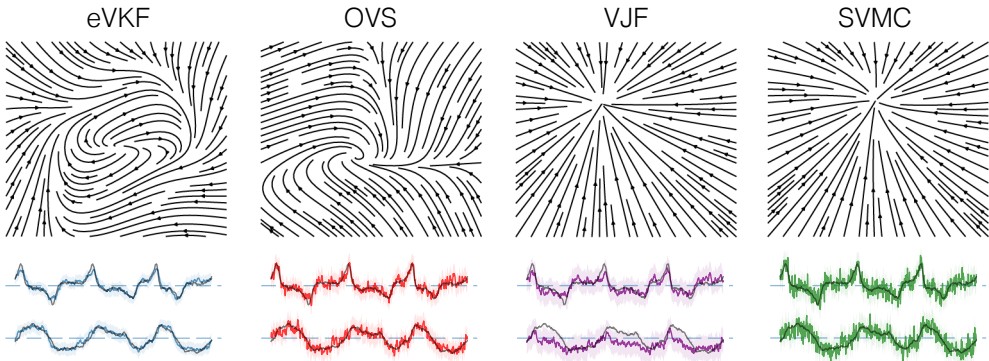

Figure 4: Learned phase portraits and inferred distribution of the latent states for the Van der Pol system with Poisson observations. For comparison, we plot the ground truth latent state in black. We see that eVKF infers much smoother latent states than the other methods.

to the filtering distribution at time $t-1$ given by $q(\mathbf{z}_{t-1}) = \mathcal{N}(\mathbf{m}_{t-1}, \mathbf{P}_{t-1})$, then the prediction step could be approximated as

$$p(\mathbf{z}_t \mid \mathbf{y}_{1:t-1}) = \mathbb{E}_{q(\mathbf{z}_{t-1})}\left[p_{\boldsymbol{\theta}}(\mathbf{z}_t \mid \mathbf{z}_{t-1})\right] \tag{44}$$

$$= \mathbb{E}_{q(\mathbf{z}_{t-1})}\left[\mathcal{N}(\mathbf{z}_t \mid \mathbf{m}_{\boldsymbol{\theta}}(\mathbf{z}_{t-1}), \mathbf{Q})\right] \tag{45}$$

$$\approx \mathbb{E}_{q(\mathbf{z}_{t-1})}\left[\mathcal{N}(\mathbf{z}_t \mid \mathbf{m}_{\boldsymbol{\theta}}(\mathbf{m}_{t-1}) + \mathbf{M}_{t-1}(\mathbf{z}_{t-1} - \mathbf{m}_{t-1}), \mathbf{Q})\right] \tag{46}$$

$$= \mathcal{N}(\mathbf{z}_t \mid \mathbf{m}_{\boldsymbol{\theta}}(\mathbf{m}_{t-1}), \mathbf{M}_{t-1}\mathbf{P}_{t-1}\mathbf{M}_{t-1}^\top + \mathbf{Q}) \tag{47}$$

$$:= \bar{q}(\mathbf{z}_t) \tag{48}$$

where $\mathbf{M}_{t-1} := \nabla \mathbf{m}_{\boldsymbol{\theta}}(\mathbf{z}_{t-1})|_{\mathbf{m}_{t-1}}$. This prediction distribution coincides exactly with the one returned by the extended Kalman filter (Särkkä, 2013), as well as the one prescribed by eVKF. Similar procedures to facilitate tractable inference in nonlinear Gaussian models is covered extensively in Kamthe et al. (2022).

# E EXPERIMENTAL DETAILS

## E.1 VAN DER POL

Data was generated according to Eq. (23) with $\gamma = 1.5$, $\tau_1 = \tau_2 = 0.1$, $\sigma = 0.1$. For the Poisson likelihood example, we can take advantage of CVI as mentioned in the main text. For this model, the expected log-likelihood has an analytical solution,

$$\mathbb{E}_{q(\mathbf{z}_t)}\log p(\mathbf{y}_t \mid \mathbf{z}_t) = \sum_n \mathbf{y}_{nt}\left(\mathbf{C}_n^\top \mathbf{m}_t - \mathbf{b}_n\right) - \Delta\exp(\mathbf{C}_n^\top \mathbf{m}_t + \tfrac{1}{2}\mathbf{C}_n^\top \mathbf{P}_t \mathbf{C}_n) \tag{49}$$

To parameterize the dynamics, $p_{\boldsymbol{\theta}}(\mathbf{z}_t \mid \mathbf{z}_{t-1})$, we use a single layer MLP with 32 hidden units and SiLU (Elfwing et al., 2018) nonlinearity. During training we use Adam (Kingma & Ba, 2014), and update the dynamics every 150 time steps. In total we use 3500 time points for training the dynamics model for all methods. For measuring the time per step as in Table 2 the experiments were run on a computer with an Intel Xeon E5-2690 CPU at 2.60 GHz.

## E.2 CONTINUOUS BERNOULLI

For the continuous Bernoulli example, we can take advantage of CVI as mentioned in the main text. For this, we require derivatives of the expected log-likelihood, which for a Gaussian likelihood, $p(\mathbf{y}_{t,n} \mid \mathbf{z}_t) = \mathcal{N}(\mathbf{C}_n^\top \mathbf{z}_t + \mathbf{b}_n, \mathbf{r}_n)$, we have that

$$\mathbb{E}_{q(\mathbf{z}_t)}\log p(\mathbf{y}_t \mid \mathbf{z}_t) = -\sum_{n,i}\frac{1}{2\mathbf{r}_n^2}\left(\mathbf{C}_{n,i}^2\mathrm{Var}(\mathbf{z}_{t,i}) + \mathbb{E}\left[\mathbf{z}_t\right]^\top \tilde{\mathbf{C}}_n\mathbb{E}\left[\mathbf{z}_t\right] - 2\mathbf{y}_{n,t}\mathbf{C}_n^\top\mathbb{E}\left[\mathbf{z}_t\right]\right) \tag{50}$$

where $\tilde{\mathbf{C}}_n = \mathbf{C}_n\mathbf{C}_n^\top$. We use Adam and update our dynamics model every 100 time steps. For this example, the synthetic data is a length 500 sequence.

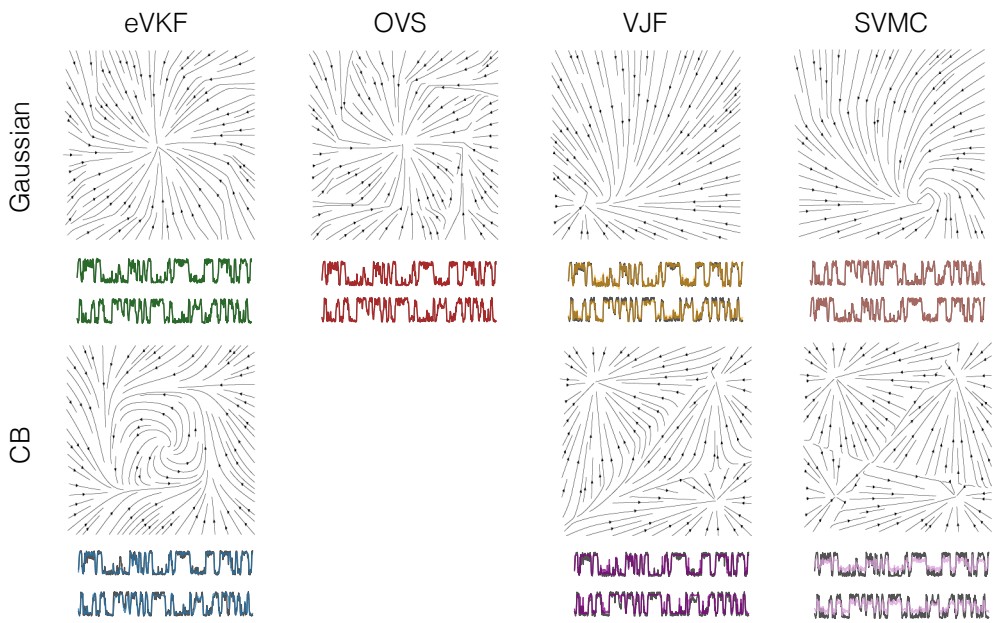

Figure 5: Learned phase portraits and inferred distribution of the latent states for the continuous Bernoulli example. The open source code for running OVS was not immediately compatible with non-Gaussian approximations. **Top**: Phase portraits and filtered latent states when approximations are constrained to be Gaussian. **Bottom**: Same as top, but for approximations constrained to be CB.

### E.3 EVKF ALGORITHM

Below we present the algorithm for using eVKF to perform inference. Instead of updating $\boldsymbol{\theta}$ every data point, we could accumulate gradients for a fixed number of steps, so that the variance of gradient steps is reduced.

---
**Algorithm 1** eVKF
---
**Input:** $\mathbf{y}_t \in \mathbb{R}^N$, $\boldsymbol{\theta}$ (dynamics parameters)
**for** each $\mathbf{y}_t$ or **until** done **do**
    $\bar{\boldsymbol{\lambda}}_t \leftarrow \mathbb{E}_{q(\mathbf{z}_{t-1})}\left[\boldsymbol{\lambda}_{\boldsymbol{\theta}}(\mathbf{z}_{t-1})\right]$          predict
    $\boldsymbol{\lambda}_t \leftarrow \text{argmax}_{\boldsymbol{\lambda}_t} \mathbb{E}_{q(\mathbf{z}_t; \boldsymbol{\lambda}_t)}\left[\log p(\mathbf{y}_t \mid \mathbf{z}_t)\right] - \mathbb{D}_{\text{KL}}\left(q(\mathbf{z}_t; \boldsymbol{\lambda}) || \bar{q}(\mathbf{z}_t; \bar{\boldsymbol{\lambda}}_t)\right)$      update
    $\ell_t \leftarrow ||\boldsymbol{\lambda}_t - \mathbb{E}_{q(\mathbf{z}_{t-1})}\left[\boldsymbol{\lambda}_{\boldsymbol{\theta}}(\mathbf{z}_{t-1})\right]||_2^2$
    $\boldsymbol{\theta}_t \leftarrow \boldsymbol{\theta}_{t-1} - \nabla_{\boldsymbol{\theta}}\ell_t$
**end for**
---

### E.4 EXAMPLE OF GAMMA DYNAMICS

We consider synthetic data where observations are conditionally Poisson and dynamic transitions are Gamma distributed, so that the state-space model description is

$$p(\mathbf{z}_{t+1} \mid \mathbf{z}_t) = \text{Gamma}(\mathbf{z}_{t+1} \mid b_0 \mathbf{f}(\mathbf{z}_t)^2, b_0 \mathbf{f}(\mathbf{z}_t)) \tag{51}$$
$$p(\mathbf{y}_t \mid \mathbf{z}_t) = \text{Poisson}(\mathbf{y}_t \mid \Delta \exp(\mathbf{C}\mathbf{z}_t + \mathbf{b})) \tag{52}$$

Similar to a standard Gaussian dynamical system, under this specification, we have that $\mathbb{E}_{p(\mathbf{z}_{t+1} \mid \mathbf{z}_t)}\left[\mathbf{z}_{t+1}\right] = \mathbf{f}(\mathbf{z}_t)$ like the gamma dynamical system presented in Schein et al. (2016), but unlike that work both $\alpha$ and $\beta$ are functions of $\mathbf{z}_t$ so that the variance is constant. We choose a variational approximation that factors as a product of Gamma distributions so that $q(\mathbf{z}_t) = \prod \text{Gamma}(\mathbf{z}_{t,i} \mid \boldsymbol{\alpha}_i, \boldsymbol{\beta}_i)$. Since we use the canonical link function, the expected log-

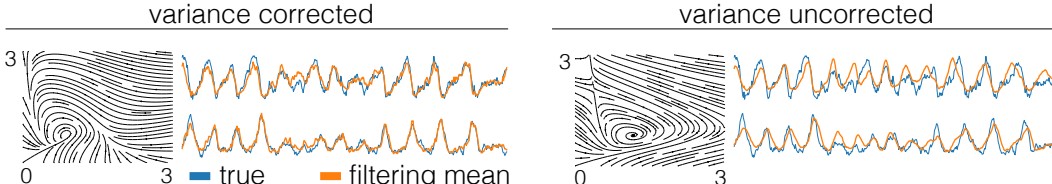

Figure 6: **On the left**: phase portrait and filtered latent states of a system with gamma distributed transitions using the prediction step variance correction. **On the right**: same as the left, but without using the prediction step correction.

likelihood can be calculated in closed form since

$$\mathbb{E}_{q(\mathbf{z}_t)} \log p(\mathbf{y}_t \mid \mathbf{z}_t) = \sum_n \left( (\mathbf{C}_n^\top \mathbf{m}_n + \mathbf{b}_n)\mathbf{y}_{t,n} - \Delta \exp(\mathbf{b}_n) \prod_{l=1}^{L} \left( 1 - \frac{\mathbf{C}_{n,l}}{\beta_{t,i}} \right)^{-\alpha_{t,i}} \right) \quad (53)$$

which allows us to use CVI for inference. We take $\mathbf{f}(\cdot)$ to be a 64 hidden unit MLP with SiLU nonlinearity and softplus output. To see how to correct for the underestimation of variance, notation will be less cluttered if our discussion is in terms of means/variances; so, let $\mathbf{z}_{t+1} \sim \bar{q}(\mathbf{z}_{t+1})$ have mean $\bar{\mathbf{m}}_{\boldsymbol{\theta},t+1}$, then the corrected variance, $\bar{\mathbf{s}}^2_{\boldsymbol{\theta},t+1}$ should be equal to the prior transition variance (i.e. $1/b_0$) plus the correction term so that $\bar{\mathbf{s}}^2_{\boldsymbol{\theta},t+1} = 1/b_0 + (\mathbf{s}_t \odot \nabla_{\boldsymbol{\theta}} \bar{\mathbf{m}}_{\boldsymbol{\theta},t+1})^2$ where $\mathbf{s}_t$ is the standard deviation of $\mathbf{z}_t \sim q(\mathbf{z}_t)$. As shown in Figure 6, without this correction, the quality of inference is noticeably worse.

