# OpenReview forum: "Real-time variational method for learning neural trajectory and its dynamics"
_ICLR.cc/2023/Conference — ICLR 2023 notable top 25%_

### Official Review · Reviewer_8dja · 2022-10-22

**Confidence:** 4
**Correctness:** 4
**Technical Novelty And Significance:** 3
**Empirical Novelty And Significance:** 3
**Recommendation:** 8

**Clarity, Quality, Novelty And Reproducibility:**

__Clarity:__
The ideas in this paper are well explained overall, however I have a few suggestions:
- Putting braces around quantities where the expectation $\mathbb{E}$ is applied would be useful. For example in the series of equations below (38), e.g. in the second line, the second expectation is applied to the quantity $\log h(z_t) + \lambda_\theta(z_{t-1})^T t(z_t) - A(\lambda_\theta(z_{t-1}))$. However the way it's written, it looks like it is only applied to $\log h(z_t)$.
- Perhaps stating explicitly that (18) implies $\log p(y_t) \geq \mathcal{L}_t(q) \geq \mathcal{M}_t(q)$ would be useful for the readers, to show the tightness of $\mathcal{L}_t(q)$ better.

__Quality:__
The quality of the paper is overall high and the maths is sound. However I spotted a few minor misprints throughout the text:
- "tracktable" in page 5 should be "tractable"
- "cVKF" in tables 2 and 3 probably meant to say "eVKF"
- "PF" in Table 1 should be "BPF" to be consistent with the text
- In section 5.1.3., "might not exactly a CB distribution" should be "might not exactly by a CB distribution"
- Proof of Theorem 2: "two ELBO's" -> "two ELBOs"
- $E$ in equation (39) should be $\mathbb{E}$ and $\bar{\lambda}$ should be $\bar{\lambda}_\theta$
- $gamma, sigma, tau$ in D.1 should be $\gamma, \sigma, \tau$

__Novelty:__
Similar ideas of framing VI as a Kalman filter type procedure exists in the literature. For example in [1, 2], a similar technique is considered when the prior (dynamics model) is Gaussian but the likelihood is non-conjugate. However, the predict step considered in this work to deal with non-Gaussian dynamics appears to be new to my knowledge. In addition, being able to learn the dynamics online using (21) is very nice and shows exciting possibilities.

[1] Khan M, Lin W. Conjugate-computation variational inference: Converting variational inference in non-conjugate models to inferences in conjugate models. Artificial Intelligence and Statistics 2017.

[2] Hamelijnck O, Wilkinson W, Loppi N, Solin A, Damoulas T. Spatio-temporal variational Gaussian processes. Advances in Neural Information Processing Systems. 2021 Dec 6.

__Reproducibility:__
Experimental details are laid out well in the appendix for reproducibility. However, details of compute used to produce the timed experiments (e.g. Table 2) would be desirable.


**Strength And Weaknesses:**

__Strengths:__

Overall the paper is well-presented and the methods are clearly explained. The ability of the model to learn nonlinear dynamics well in an online setting presents exciting possibilities.

__Weaknesses:__

While the ideas presented in this paper are neat, the need to correct the filtering distribution in (23) is a slight sore point as it doesn't naturally follow from the derivations. I believe that this may be possible to overcome by minimising (12) directly instead of using the upper bound (11), where (9) can be computed approximately, e.g. by linearising the dynamics model $m_\theta$ (as done for example in [1]).

[1] Kamthe S, Takao S, Mohamed S, Deisenroth M. Iterative State Estimation in Non-linear Dynamical Systems Using Approximate Expectation Propagation. Transactions on Machine Learning Research. 2022.

**Summary Of The Paper:**

In this paper, the authors consider state estimation problems in the setting where the underlying dynamics model is unknown. They introduce the exponential family variational Kalman filter (eVKF), which is a variational inference scheme for state-space systems that consists of a prediction step and an update step, much like the Kalman filter. The approximate posteriors are assumed to be in the exponential family to make computations tractable. In addition, the latent models can be learnt via an EM algorithm in a tractable fashion. They prove that the two-step predict-update inference procedure gives tighter bounds compared to directly finding an approximate filtering distribution using variational inference and show that it works well on several experiments.

**Summary Of The Review:**

The paper is of good quality and the methods presented are backed up by rigorous mathematical arguments. Aside from minor issues and the fact that similar ideas exist in the literature, the ideas and experiments are executed well here. The paper definitely deserves to be accepted at ICLR.

---

### Official Review · Reviewer_b7sM · 2022-10-23

**Confidence:** 4
**Correctness:** 3
**Technical Novelty And Significance:** 2
**Empirical Novelty And Significance:** 3
**Recommendation:** 6

**Clarity, Quality, Novelty And Reproducibility:**

The paper is overall clear and easy to follow. The idea is effective to the problem but short of further discussions (please see Weakness for more details).

**Strength And Weaknesses:**

Strength

1. The specific exponential family is adopted to ease the filtering posterior distribution and latent dynamic learning.
2. The guaranteed tightness ELBO comparing straightforward variational update.
3. The consistent better performance.

Weakness

1. The specific exponential family, i.e., with constant base measure and minimal, is used without further discussions on the possible side effects. The constant base measure may not a strong restricted condition because many well known distributions are within this class, but what effects from the minimal?  especially for neural trajectory modeling and prediction?

2. There are various works about variational kalman filtering in the literature like [1, 2]. Why these methods are not compared in experiments and even not discussed in related works?

3. eVKF is claimed to be a 'novel variational inference scheme' but it is just a special case of VKF under a specific exponential family. It cannot be considered as a novel one.



[1] Das, Niladri, Jed A. Duersch, and Thomas A. Catanach. "Variational Kalman Filtering with Hinf-Based Correction for Robust Bayesian Learning in High Dimensions." arXiv preprint arXiv:2204.13089 (2022).

[2] Huang, Yulong, Yonggang Zhang, Peng Shi, and Jonathon Chambers. "Variational adaptive Kalman filter with Gaussian-inverse-Wishart mixture distribution." IEEE Transactions on Automatic Control 66, no. 4 (2020): 1786-1793.





**Summary Of The Paper:**

This paper proposes an exponential family variational Kalman filter (eVKF) to model and predict the neural trajectory. The basic idea is to restrict the filtering posterior and latent dynamics to a special exponential family with constant base measure and minimal. The restricted distribution form could lead to efficient filtering and update. The proposed two step procedure is proofed to be able to bring tightness of ELBO. The empirical study shows the effectiveness of the proposed idea.

**Summary Of The Review:**

This paper is with clear motivation and the idea is reasonable with guarantee. The experiments show the effectiveness but without comparing with related works like existing variational Kalman filter.

---

### Official Review · Reviewer_6sm4 · 2022-10-23

**Confidence:** 3
**Correctness:** 3
**Technical Novelty And Significance:** 3
**Empirical Novelty And Significance:** 3
**Recommendation:** 6

**Clarity, Quality, Novelty And Reproducibility:**

This paper offers an approximate solution to a problem, trying to make things tractable under special assumptions. An alternative comparison would be compared to more flexible models. For example, how does the proposed approach compare to KalmanNet (Revach, Guy, et al, 2022)?

Revach, Guy, et al. "KalmanNet: Neural network aided Kalman filtering for partially known dynamics." IEEE Transactions on Signal Processing 70 (2022): 1532-1547.

**Strength And Weaknesses:**

Pros:

This paper is very well-written and presented in an organized, self-contained way.

This paper offers an approximate solution that extends to non-Gaussian non-linear models even in latent spaces, while still staying relatively tractable.

The evaluation of eVKF with real-world data and comparisons to GPFA is helpful.

Cons:

Although the online real-time learning aspect is well-motivated, it is unclear what factors contribute to achieving it, in making things possible. What are some of the bottlenecks of previous methods? Is backpropagation really bad for the real-time setting considered? The claim of real-time also needs to consider the data rate. What is the data rate per sample if the processing time is about 10ms per sample?

**Summary Of The Paper:**

This paper presents a variational approach for filtering and estimating the state parameters in an online manner.  In particular, the prediction distribution and filtering distribution are assumed to be within the constant base measure exponential family. With this assumption, many computations benefit from the conjugacy, such as the natural parameter of variational prediction distribution is in closed form, and the learning of parameters for the dynamic model can be simplified. The proposed approach also leads to a tighter lower bound than previous works. Experimental results include both synthetic data, using appropriate performance measures, and a compelling real-world use case.

**Summary Of The Review:**

The writing of the paper is high-quality and the analysis seems very interesting.

---

### Official Review · Reviewer_Hbu9 · 2022-10-24

**Confidence:** 4
**Correctness:** 3
**Technical Novelty And Significance:** 4
**Empirical Novelty And Significance:** 3
**Recommendation:** 8

**Clarity, Quality, Novelty And Reproducibility:**

- The paper is well-written and the claims are mostly well-supported.
- The technical novelty is given and the algorithm is fairly easy to implement and as such should be easily reproducible.
- The results look convincing

**Strength And Weaknesses:**

Filtering and smoothing are probably one of the most applicable topics for time series data. Therefore, this paper which proposes a new filtering algorithm is of high interest to the community. The developed approach is interesting and novel to me. The algorithm development is easy to follow. The experimental evaluation is carried out nicely and a comparison to standard baselines and some newer state-of-the-art algorithms are given.
One thing that I find problematic is the use of the upper bound in the prediction step. This upper bound as described in Sec 3.6. leads to an underestimation of the variance, which is heuristically corrected. This correction step, however, does not follow a principled derivation and as such, it remains unclear if it is sensible to use in every setup. Also note, that in my version the discussion section Appendix D.4 is missing.
Otherwise, I think this is a solid piece of work.

A question to the authors is: When using the standard ELBO objective in the prediction step, e.g., with stochastic variational inference, is the problem of underestimating the variance still present?

As an aside: I am not so happy with the name eVKF as the filter does not assume the usual assumption that Kalman-type filters have (some form of linearity and Gaussianity). Maybe, it is a good idea to think about a slightly different name.

**Summary Of The Paper:**

The paper considers the problem of optimal filtering and simultaneous parameter estimation for discrete-time systems with exponential family latent dynamics. The authors propose a new variational inference algorithm, by splitting the prediction and update step in the filtering algorithm. The variational approximation for the prediction step is based on an upper-bound minimization and the update step is based on a lower-bound (ELBO) maximization. Parameter learning is performed using variational expectation maximization, which yields an easy objective under the exponential family assumptions.
The algorithm is empirically evaluated on three synthetic filtering problems and one real-data example.

**Summary Of The Review:**

The paper discusses a new filtering and parameter learning algorithm using variational inference. The paper has convincing results, however, a heuristic is used to improve the results, where it is not very clear to me how and when it should be applied for setups different from those discussed in the paper.

---

### Decision · Program_Chairs · 2023-01-20

**Decision:**

Accept: notable-top-25%

**Justification For Why Not Higher Score:**

eVKF is claimed to be a 'novel variational inference scheme' but it is a special case of VKF under a specific exponential family. Thus the originality is questioned. Also, there are various works about variational kalman filtering in the literature. These methods are not compared in experiments and not discussed in related works.



**Justification For Why Not Lower Score:**

The novelty is significant. The ability of the model to learn nonlinear dynamics well in an online setting presents exciting possibilities.


**Metareview: Summary, Strengths And Weaknesses:**

This paper proposes an exponential family variational Kalman filter (eVKF) to model and predict the neural trajectory. The basic idea is to restrict the filtering posterior and latent dynamics to a special exponential family with constant base measure and minimal. The restricted distribution form could lead to efficient filtering and update. The proposed two step procedure is proofed to be able to bring tightness of ELBO. The empirical study shows the effectiveness of the proposed idea. This paper is very well-written and presented in an organized, self-contained way. It offers an approximate solution that extends to non-Gaussian non-linear models even in latent spaces, while still staying relatively tractable.The experimental results on the real-world dataset are convincing.

The novelty is significant. The ability of the model to learn nonlinear dynamics well in an online setting presents exciting possibilities.


**Note From Pc:**

if the above contains the word "oral" or "spotlight" please see: "oral" presentation means -> notable-top-5% and "spotlight" means -> notable-top-25%. As stated in our emails, we are disassociating presentation type from AC recommendations